

# AresB-Net: accurate residual binarized neural networks using shortcut concatenation and shuffled grouped convolution

HyunJin Kim

School of Electronics and Electrical Engineering, Dankook University, Yongin, South Korea

## ABSTRACT

This article proposes a novel network model to achieve better accurate residual binarized convolutional neural networks (CNNs), denoted as AresB-Net. Even though residual CNNs enhance the classification accuracy of binarized neural networks with increasing feature resolution, the degraded classification accuracy is still the primary concern compared with real-valued residual CNNs. AresB-Net consists of novel basic blocks to amortize the severe error from the binarization, suggesting a well-balanced pyramid structure without downsampling convolution. In each basic block, the shortcut is added to the convolution output and then concatenated, and then the expanded channels are shuffled for the next grouped convolution. In the downsampling when *stride* >1, our model adopts only the max-pooling layer for generating low-cost shortcut. This structure facilitates the feature reuse from the previous layers, thus alleviating the error from the binarized convolution and increasing the classification accuracy with reduced computational costs and small weight storage requirements. Despite low hardware costs from the binarized computations, the proposed model achieves remarkable classification accuracies on the CIFAR and ImageNet datasets.

## INTRODUCTION

Neural networks have achieved significant performance enhancements in many fields, including computer vision, speech recognition, and natural language processing, etc. Notably, convolutional neural networks (CNNs) have shown many outstanding performances in the field of computer vision. Even though it is possible to implement simple CNNs such as LeNet-5 (*LeCun et al., 1998*) on embedded devices, more sophisticated CNNs require high computational costs and large weight storage requirements, which prevent them from being adopted in lightweight cost-efficient systems. Various studies try to reduce memory requirements and power consumption at the expense of the appropriate performance degradation. The network quantization sacrifices the precision of model parameters and approximates the operations in neural networks to achieve small memory requirements and power consumption (*Wu et al., 2016*; *Zhou et al., 2017*).

Corresponding author
HyunJin Kim,
hyunjin2.kim@gmail.com

Notably, early approaches for the binarized neural network (BNN) models in *Courbariaux, Bengio & David (2015)*, *Courbariaux et al. (2016)* and *Rastegari et al. (2016)* quantize weights or activations into {+1, −1}, which replaces floating-point multiplications with binary bitwise operations, thus approximating the floating-point multiply-accumulate operation using bitwise XNOR and bit counting operations. Besides, the quantized binary weights can reduce weight storage requirements, which makes BNNs a highly appealing method for implementing CNNs on embedded systems and programmable devices (*Guo, 2018*; *Zhou, Redkar & Huang, 2017*; *Yi, Xiao & Yongjie, 2018*; *Liang et al., 2018*). Despite many benefits above, the low precision of the binarized operations in BNNs degrades the classification ability on modern CNNs, thus limiting their applications. Improved BNN structures have been developed for reducing the gap of the classification accuracy degraded over real-valued CNNs (*Lin, Zhao & Pan, 2017*; *Liang et al., 2018*; *Liu et al., 2018*; *He et al., 2018*; *Zhuang et al., 2019*; *Shen et al., 2019*; *Chakraborty et al., 2019*; *Bethge et al., 2019*; *Phan et al., 2020*; *Bethge et al., 2020*; *Liu et al., 2020*). Besides, new training methods and optimizing tricks for BNNs have been researched for obtaining better classification accuracy (*Alizadeh et al., 2018*; *Bulat & Tzimiropoulos, 2019*; *Zhu, Dong & Su, 2019*; *Wang et al., 2019*; *Hubara et al., 2017*; *Ghasemzadeh, Samragh & Koushanfar, 2018*; *Gu et al., 2019*; *Helwegen et al., 2019*; *Ding et al., 2019*; *Martinez et al., 2020*). However, there are still significant accuracy drops compared with the baseline floating-point models. The insufficient feature *resolution* from the binarized convolution layer can be compensated using real-valued shortcuts in *Liu et al. (2018)*, thus making a noticeable advance in increasing the classification accuracy. However, it is concerned that the stacking structure of residual convolution layers accumulates errors from each binarized layer, which can limit the performance of residual CNNs. The feature reuse of BNNs in *Bethge et al. (2019, 2020)* concatenates shortcuts to expand output channel, making features from the shortcut pass to the next block. These existing methods in *Liu et al. (2018)* and *Bethge et al. (2019)* adopt point-wise convolutions in the channel expansion, which can increase computational costs in BNNs.

Our approach combines the advantages of the feature resolution enhancement and feature reuse schemes, eliminating the convolutions in the channel expansion. The proposed network model called *AresB-Net* is developed to consider these motivations. The basic block connects the real-valued shortcut per each binarized convolution layer by adding the shortcut and concatenating it to output channels. Two different kinds of shortcuts are mixed for expanding channels. In the downsampling, only the max-pooling layer is used for obtaining the shortcut. Then, the doubled expanded channels are shuffled and split for the grouped convolution, so that computational costs are reduced without downsampling $1 \times 1$ convolution compared with baseline BNNs. The ratio of unfiltered features is naturally maintained in the shuffled channels. Similar to the baseline residual CNNs of *He et al. (2016)*, the proposed basic block are easily stacked to create the pyramid structure of CNNs. In experiments, with well-known data augmentation and regularization techniques, this novel BNN structure provides 91.90% and 73.01% Top-1 accuracies with the 18-layered models on the CIFAR-10 and CIFAR-100 datasets

(*Krizhevsky, Nair & Hinton, 2014*) and 78.15% Top-5 accuracies on the ImageNet dataset (*Russakovsky et al., 2015*).

In the following, we introduce several related works and explain our motivation for the proposed structure. Then, the details of the proposed BNN structure are described. Finally, experimental results show the classification accuracy and computational cost analysis.

# RELATED WORK

## Residual CNNs

Between stacked layers in a network model, the skip connection called *shortcut* can jump over one or several non-linear activations, so it is summed to the other layer output. Thus, the shortcut contains unfiltered features from previous layers, which enable the residual networks to achieve fast training speed with the reduced impact of vanishing gradients and obtain the benefits from ensemble effects (*Veit, Wilber & Belongie, 2016*). In general, there are two different shortcut summing schemes in residual CNNs; (1) adding the shortcut to each channel without changing the number of output channels (*He et al., 2016*): (2) concatenating the shortcut for expanding channels (*Huang et al., 2017*; *Zhang et al., 2018*). Whereas adding the shortcut to the layer output can *dilute* the unfiltered features, the channel expansion requires computational costs in the point-wise convolution between channels. In our study, it is motivated that both features from the two summing schemes above can be mixed in each block, thus expanding channels without increasing computational costs.

Several network models using the grouped convolution adopt the residual structure for summing the shortcut to their basic block. Especially, the shortcut is summed to the shuffled channels for the grouped convolutions in *Zhang et al. (2018)*. Besides, MobileNetv2 (*Sandler et al., 2018*) introduces the inverse residual structure containing depth-wise convolutions. The works in *Zhang et al. (2018)* and *Sandler et al. (2018)* prove that summing the shortcut to the grouped convolution output obtains considerable classification accuracy with decreased computational costs. In our proposed block, whereas the grouped convolution reduces the computational costs for expanded input channels, the difference from *Zhang et al. (2018)* and *Sandler et al. (2018)* is that the features shuffled from two different residual shortcuts are used in each group.

## Binarized CNNs

As the complexity of neural networks increases, large memory requirements and high computational costs are significant burdens when applying CNNs on edge computing devices. Notably, increasing multiplications require high power consumptions that embedded devices cannot accept. BNNs quantize weights (*Courbariaux, Bengio & David, 2015*) or both weights and activations (*Hubara et al., 2016*, *2017*; *Rastegari et al., 2016*) of neural networks into $\{-1, +1\}$. The analysis of the inference step in *Rastegari et al. (2016)* shows $\approx 32 \times$ memory saving and $\approx 58 \times$ computation speedup, thus making BNNs an appealing neural network scheme in embedded systems. However, when applied directly to the baseline real-valued neural network model, errors from the binarization degrade the classification accuracy. In *Courbariaux, Bengio & David (2015)*, the training

scheme for BNNs is introduced. In XNOR-Net (*Rastegari et al., 2016*), the binarized network structure and convolution with the deterministic scaling factor make significant classification improvements, which are verified empirically in the residual CNN of *He et al. (2016)*. Beyond these early works of BNNs, new basic blocks that utilize residual networks have been developed. Especially, in *Liu et al. (2018)*, each basic residual block has the real-valued shortcut for skipping only one non-linear activation, which has been adopted in other BNNs. The grouped convolutions are applied for adopting the binarized depth-wise separable convolution in *He et al. (2018)* and *Phan et al. (2020)*. In *Bethge et al. (2019, 2020)*, shortcut is concatenated to expand channels in dense neural networks.

From these previous works, we conclude that the residual binarized basic blocks have layered structures different from the real-valued baselines. Our BNN is based on the shuffled grouped convolution and combines different shortcuts in the residual binarized blocks, which are discriminated from other residual BNNs. In *Bulat & Tzimiropoulos (2019)* and *Liu et al. (2020)*, trainable parameters are used in the activation and scaling. In *Alizadeh et al. (2018)*, *Zhu, Dong & Su (2019)*, *Wang et al. (2019)*, *Hubara et al. (2017)*, *Ghasemzadeh, Samragh & Koushanfar (2018)*, *Gu et al. (2019)*; *Helwegen et al. (2019)*, *Ding et al. (2019)* and *Martinez et al. (2020)*, the training and optimization techniques for BNNs have been studied. Even though trainable parameters and optimizing techniques can be useful in increasing classification accuracy, our method does not consider any other specific trainable parameters and training techniques.

# ARESB-NET: ACCURATE RESIDUAL BINARIZED NEURAL NETWORK MODEL

The proposed AresB-Net model contains novel basic blocks using residual shortcuts, expanding channels by adding and concatenating shortcuts. This basic blocks can be stacked using a pyramid structure. Most CNN structures reduce the width and height of feature maps and increase the number of channels when they encounter a layer with downsampling (*stride* > 1). Because the baseline residual networks (*He et al., 2016*) and XNOR ResNet (*Rastegari et al., 2016*) simply adopt *stride* = 2 and double channels in the downsampling, the AresB-Net also follows this pyramidal method using a factor of 2. In this downsampling, whereas the width and height of features are reduced in half, the number of channels are doubled. Therefore, the amount of computation on each layer is kept similar.

This section explains the basic block for this pyramid structure and its binarization of features and weights. Then, the model structure stacking the basic blocks is described. Finally, we summarize the training method for the proposed AresB-Net.

## Proposed basic block

Figure 1 shows the proposed basic block. Two kinds of shortcut summing for expanding channels are illustrated: (1) adding the shortcut from the first batch normalization (BN) to the output of the second BN layer; (2) concatenating the shortcut from the first BN layer to the output channels. The BinConv3×3 layer stands for the binarized convolution layer with 3 × 3 sized filter. This concatenated shortcut does not go through

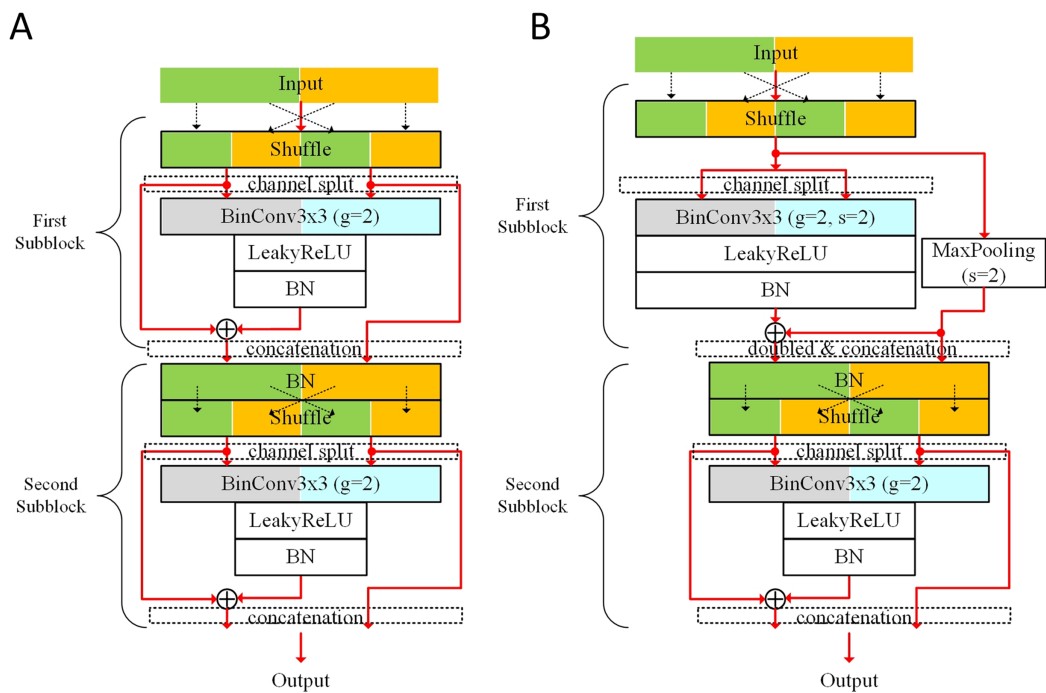

**Figure 1 Proposed basic blocks: (A) basic block for *stride* = 1; (B) basic block for *stride* = 2.**

the BinConv3×3 layer, expanding the output channels without additional computational costs. When *stride* = 2, the number of output channels from the BinConv3×3 layer is doubled, and 3 × 3 max-pooling layer is used to make the shortcut. Therefore, the number of output channels from the first subblock is doubled.

Before forwarding input features into each BinConv3×3 layer, the input channels are shuffled and then split for the grouped convolution. This basic block uses the shuffle layer described in *Zhang et al. (2018)*. The input channels contain the features generated from different types of shortcuts. This shuffling can mix the features from the channels and divide them into two groups (denoted as *g* = 2). In this shuffling, the information crossing input channels is mixed in the next grouped convolution, which is helpful for removing real-valued 1 × 1 convolution like *Zhang et al. (2018)*. This scheme manages the ratio of the reused unfiltered features from the previous layers. As the input features produced from a subblock go though other subblocks, the ratio of unfiltered features from the subblock decreases. When half of the features produced from a subblock are concatenated to the output channels of the next subblock, the features are not unfiltered in the next subblock. We denote the number of features from a subblock by num(**I**). In the output channels of the next subblock, $\frac{\text{num}(\mathbf{I})}{2}$ features are unfiltered. As the features go through *k* subblocks, $\frac{\text{num}(\mathbf{I})}{2^k}$ features are also unfiltered.

The structure with the BN layer before the binary activation follows the baseline work in *He et al. (2016)* and *Rastegari et al. (2016)*, where learnable shifting and biasing parameters γ and β for each channel transform values for determining which value is binarized into −1 or +1 in the binarized activation. Compared with ResNet (*He et al., 2016*)

and XNOR ResNet (*Rastegari et al., 2016*) models, the proposed model contains doubled shifting and biasing parameters, which could give more optimizable points in training.

Each BinConv3×3 layer consists of the deterministic binarized activation and convolution. Let us assume that term $\mathbf{I} \in \mathbb{R}^{c \times w_{in} \times h_{in}}$ denotes the input tensor of a basic block, where $c$, $w_{in}$, and $h_{in}$ mean the number of input channels, and the width and height of a feature map, respectively.

In the first subblock, the deterministic binarized activation $sign(\mathbf{I})$ rounds $\mathbf{I}$ into $\{-1, +1\}^{c \times w_{in} \times h_{in}}$. When the weight filter for each input channel has the width of $w$ and height of $h$, the real-valued weight filter for an output channel is denoted as $\mathbf{W} \in \mathbb{R}^{n = \frac{1}{2} c \times w \times h}$, where $w \leq w_{in}$ and $h \leq h_{in}$. In the BinConv3×3 layer, $w = 3$ and $h = 3$, respectively. Because the BinConv3×3 layer performs the grouped convolution ($g = 2$), $\frac{1}{2} c$ input channels are adopted in each group.

Depending on the *stride*, the numbers of output channels from the first subblock are different. As shown in Fig. 1, whereas the first BinConv3×3 layer of *stride* = 1 has $\frac{1}{2} c$ output channels, that of *stride* = 2 has $c$ output channels. When *stride* = 1, $c$ input channels for the second subblock are produced by concatenating the shortcut from $\frac{1}{2} c$ shuffled input channels. On the other hand, for *stride* = 2, $2c$ input channels are produced for the second subblock, where $c$ channels from the max-pooling layer are concatenated to produce $2c$ channels. By applying *stride* = 2, the width and height of the feature from the subblock are $\frac{w_{in}}{2}$ and $\frac{h_{in}}{2}$, respectively. In the second subblock, the numbers of input and output channels are the same. Therefore, the output tensor sizes of *stride* = 1 and *stride* = 2 are $c \times w_{in} \times h_{in}$ and $2c \times \frac{w_{in}}{2} \times \frac{h_{in}}{2}$, respectively.

In a group of the BinConv3×3 layer, when the numbers of input and output channels are the same as $\frac{c}{2}$, the number of parameters for the group convolution can be $\frac{1}{4} c^2 \times w \times h$. The total number of parameters for two groups can be $\frac{1}{2} c^2 \times w \times h$. When the number of output channels in a group is doubled as $c$ in the first subblock for *stride* = 2, the total number of parameters for two groups can be $c^2 \times w \times h$. Table 1 summarizes the number of parameters used in weight filters denoted as weight size and output tensor sizes in basic blocks.

## Binarization

When binarizing $\mathbf{W}$ with $sign(\mathbf{W})$, only the binary weight $\mathbf{B} \in \{-1, +1\}^{n = \frac{1}{2} c \times w \times h}$ for each input channel is used in the inference. In the binarized activation and weights, function $sign(x)$ is defined as:

$$x \in \{\mathbf{I}, \mathbf{W}\}, \ sign(x) = \begin{cases} +1 & \text{if } x \geq 0 \\ -1 & \text{else} \end{cases} \tag{1}$$

Thus, the binarized convolution output is approximated as:

$$\mathbf{I} \times \mathbf{W} \approx (sign(\mathbf{I})\{ \circledast sign(\mathbf{W}))\odot\alpha \tag{2}$$

where $\alpha$ denotes the scaling factor for weights. As shown in *Rastegari et al. (2016)*, the scaling factor is $\frac{1}{n}$, where $n = \frac{1}{2} c \times w \times h$. Symbols $\circledast$ and $\odot$ mean the convolution using

**Table 1 Summary of weight and output tensor sizes in basic blocks.**

| Block type | Input tensor size | Subblock | Weight size[a] | Output tensor size |
|---|---|---|---|---|
| *stride* = 1 | $c \times w_{in} \times h_{in}$ | First | $\frac{1}{4}c^2 \times w \times h$ | $c \times w_{in} \times h_{in}$ |
| | | Second | $\frac{1}{4}c^2 \times w \times h$ | $c \times w_{in} \times h_{in}$ |
| *stride* = 2 | | First | $\frac{1}{2}c^2 \times w \times h$ | $2c \times \frac{w_{in}}{2} \times \frac{h_{in}}{2}$ |
| | | Second | $c^2 \times w \times h$ | $2c \times \frac{w_{in}}{2} \times \frac{h_{in}}{2}$ |

**Note:**
[a] Weight size denotes the number of weight filter's parameters.

bitwise XNOR & bit-counting operations and element-wise scalar multiplication, respectively. After binarizing weights, the multiplication with the binarized activations is approximated using the bitwise XNOR operation. Because each operand consists of one bit, the bitwise XNOR operation can perform the parallel multiplication in a bit-level. The accumulation operation can be replaced by the bit-counting operation. In Eq. (2), the binarized convolution only adopts the deterministically scaled weights by $\frac{1}{n}$. Calculating the element-wise scaling factor matrix $\mathbf{K}$ for $\mathbf{I} \approx sign(\mathbf{I}) \odot \mathbf{K}$ in the inference (*Rastegari et al., 2016*) is a significant burden in lightweight BNNs, as described in *Bulat & Tzimiropoulos (2019)*. Instead, in our design, this convolution output is adjusted by the learnable affine parameters in the following BN layer.

The erroneous binarized convolution can increase unexpected *dying ReLU* problems. Several existing works adopted the learnable leaky activation functions (*Gu et al., 2019*; *Phan et al., 2020*; *Martinez et al., 2020*). The leaky activation function allows small negative values when input is less than zero. Therefore, we evaluated whether the leaky activation function can be suitable for the proposed model. Evaluations were performed by changing the activation function to the LeakyReLU (*Maas, Hannun & Ng, 2013*), ReLU (*Nair & Hinton, 2010*), parametric ReLU (PReLU) (*He et al., 2015*) in the AresB-18 model on the CIFAR-100 dataset. In this evaluation, the negative slope of the LeakyReLU was fixed as 0.01. Top-1 final test accuracies with the LeakyReLU, ReLU, PReLU were 73.01%, 71.94%, 71.23%, respectively. The evaluation result using the LeakyReLU outperformed other activation functions, so that we decided that the binarized convolution output passed through the LeakyReLU layer.

The first BN layer in the second subblock normalizes all features from the first subblock, where the unfiltered features from previous blocks can be adjusted in this BN layer. On the other hand, the first subblock does not have the BN as the first layer. We think that if all features pass through the BN layer in each subblock, errors from the repeated normalization could have negative effects, which produced 72.95% Top-1 final test accuracy on the CIFAR-100 dataset in our experiments. When the first subblock did not adopt the BN layer, 73.01% Top-1 final test accuracy was obtained, so the difference was not significant. However, additional BN layer increased computational costs, so that it was expected that there was no need to insert that layer. Therefore, we determine that a basic block has this BN layer every two subblocks in AresB-Net.

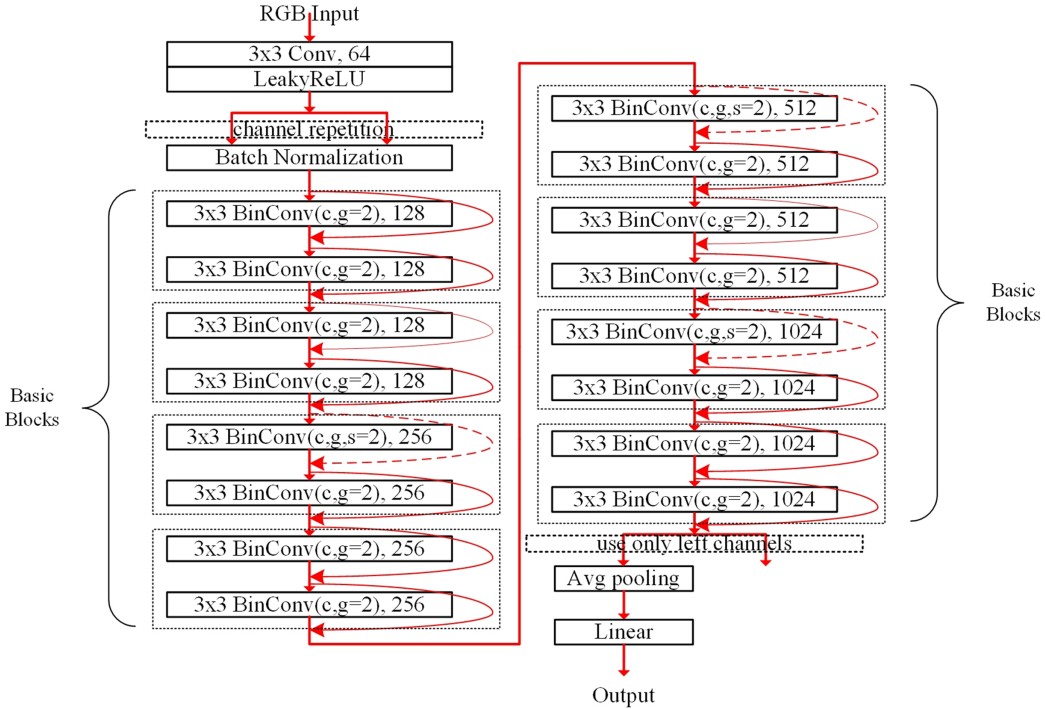

**Figure 2 Pyramid structured model stacking eight basic blocks denoted as the AresB-18 model for the CIFAR dataset.** The number in the box and term $g = 2$ denote the numbers of the output channels and groups in the convolution.

## Model structure

Figure 2 describes the pyramid structure containing eight basic blocks for the CIFAR dataset. The dotted box denotes the basic block that consists of two subblocks. Following the baseline residual networks in *He et al. (2016)*, the basic blocks are stacked. But the channels are extended without increasing computational costs using the grouped convolution in the convolution layers. As shown in Fig. 2, the first convolution layer performs the real-valued convolution using image pixel input data from RGB channels; other convolutions are binarized. After copying the output of the LeakyReLU layer (*Maas, Hannun & Ng, 2013*), the BN layer adjusts the features. In the first subblock of the first basic block, the channels are not shuffled because the repeated channels contain the same features. Except for the first real-valued convolution layer, the subblock having a convolution layer is connected with its shortcut, which is denoted as the rounded red arrow. The dotted round arrow indicates that the features from the previous basic block pass through the max-pooling layer with *stride* = 2, thus making the number of channels doubled per two basic blocks.

The second subblock of the final basic block does not concatenate the shortcut, so that the output channels are obtained just by adding the shortcut from the first BN layer to the output of the second BN layer, so that left channels are used in average pooling in Fig. 2. Therefore, the number of output channels is the same with that of the baseline residual networks in *He et al. (2016)*. After performing the average pooling, the real-valued

linear fully-connected layer makes the image classification result. The model structure for the ImageNet dataset has the same concept for stacking the basic blocks. The detail modification for the ImageNet dataset is described in the section of the experimental results and analysis.

Our model structure expands channels compared with the baseline residual CNNs (*He et al., 2016*). Because the grouped convolution is applied to the channel expansion for concatenating features, there is no increase in computational costs. When *stride* = 2, the max-pooling layer obtains the downsampled real-valued features to be concatenated. Our method does not adopt $1 \times 1$ binarized convolutions in the downsampled shortcut to reduce storage size and computational costs. In *Rastegari et al. (2016)* and *Liu et al. (2018)*, the downsampled shortcut adopts $1 \times 1$ real-valued convolutions to preserve the information between blocks. However, we concern that the real-valued convolutions in the downsampled shortcut reduce the degree of the binarization in BNNs, which increases the memory requirements for storing weights. Besides, it is assured that the computational costs of the max-pooling layer are much smaller than those of $1 \times 1$ real-valued convolution.

## Training of proposed BNNs

When training our proposed AresB-Net model, weights are binarized in the forward pass and backward propagation, following the training method described in *Rastegari et al. (2016)*. In the forwarding pass, the binarized activation and convolution are performed based on Eqs. (1) and (2). When updating parameters, real values are maintained to keep the tiny change in parameters. Especially, in the backpropagation, the derivative of the binary activation using *sign*() function should be approximated because the ideal derivative of *sign*() function is the delta function. Even though it is known that more sophisticate approximated derivatives such as *Liu et al. (2018)* can provide better results, we adopt the baseline *straight-through-estimator* in *Courbariaux et al. (2016)* for the training.

## EXPERIMENTAL RESULTS AND ANALYSIS

Our proposed model was trained and then tested in terms of image classification accuracy. In this experiment, the CIFAR (*Krizhevsky & Hinton, 2009*) and ImageNet (*Russakovsky et al., 2015*) datasets were adopted, where different experimental environments were setup. For apple-to-apple comparison, we adopted commonly used optimizers such as SGD (stochastic gradient descent) (*Bottou, 2010*) and ADAM (*Kingma & Ba, 2014*) optimizer in this training. Even though we did not apply the specific training scheme, it was concluded that our model could achieve significant accuracy enhancements in residual BNNs.

### Experiments on CIFAR dataset

In the training and testing, CIFAR-10 and CIFAR-100 datasets were used. The CIFAR dataset consists of 60,000 $32 \times 32$ colour images, where 50,000 and 10,000 images are used in the training and test, respectively. Whereas the CIFAR-10 dataset is classified into 10 different classes, the CIFAR-100 dataset has 100 classes containing 600 images for each

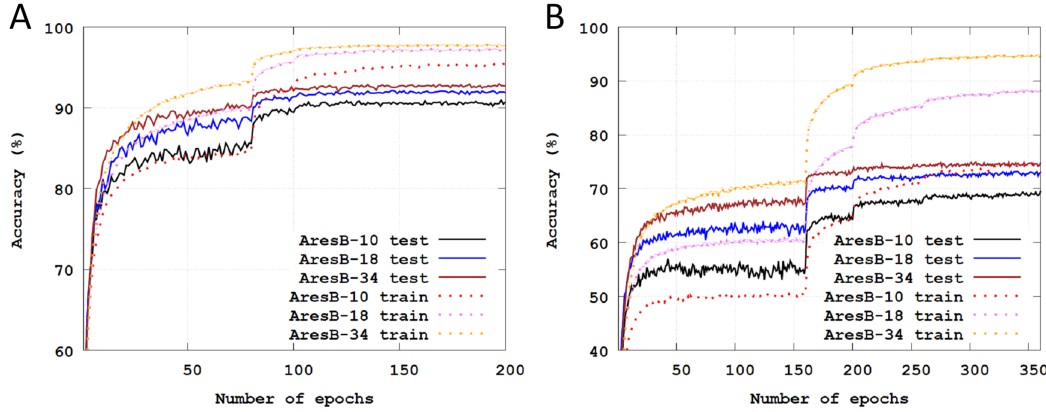

**Figure 3** Training and test classification accuracies across training epochs: (A) training and test Top-1 accuracies with the AresB-10, AresB-18, and AresB-34 models on the CIFAR-10 dataset; (B) training and test Top-1 accuracies with the AresB-10, AresB-18, and AresB-34 models on the CIFAR-100 dataset.

class. As the number of classes increased, it is noted that more sophisticate classification was required.

In our experiments, three different models denoted as the AresB-10, AresB-18, and AresB-34 models, were trained and then tested, where the AresB-10, AresB-18, and AresB-34 models stacked 4, 8, and 16 basic blocks, respectively. The structure of the AresB-18 model is described in Fig. 2. We used SGD optimizer with momentum=0.9 and weight decay=1$e$−5. Unlike *Lin, Zhao & Pan (2017)*, *Liu et al. (2018)*, *Bulat & Tzimiropoulos (2019)* and *Martinez et al. (2020)*, the real-valued pretrained weights for initializing the models were not adopted, thus starting the training from randomly initialized weights.

For the data augmentation for input images, 32 × 32 input image was cropped and horizontally flipped randomly from 40 × 40 padded image (*padding* = 4). This random crop and random horizontal flip were applied to the evaluations of the AresB-Net and other counterparts. Then, the random erasing introduced in *Zhong et al. (2017)* was applied in training. However, the data augmentation above was not applied in the testing. The random erasing was only adopted in the training of the AresB-Net, so that other counterparts did not use this augmentation technique. For the CIFAR-10 dataset, we ran the training for 200 epochs with a batch size of 256. The learning rate started at 0.1 and was decayed by multiplying 0.2 at (80, 100, 130, 160) epochs. For the CIFAR-100 dataset, the training was performed for 360 epochs with a batch size of 256, where the learning rate started at 0.1 and was decayed by multiplying 0.2 at (160, 200, 260, 320) epochs. For the CIFAR-100 dataset, the dropout (*Srivastava et al., 2014*) layer was placed just before the fully-connected layer.

Figure 3 illustrates Top-1 classification accuracies across training epochs on the CIFAR datasets. The solid and dashed lines represent test and training accuracies, respectively. In our experiments, the final test accuracy drops from full-precision models were ranged in 0.91–2.60%. As the number of stacked blocks increased, there were

**Table 2 Summary of test accuracies (%) on CIFAR datasets.**

| Dataset | Model | Top-1 | Top-5 | FP Top-1[a] | FP Top-5[a] | Top-1 gap | Top-5 gap |
|---------|-------|-------|-------|-------------|-------------|-----------|-----------|
| CIFAR-10[a] | AresB-10 | 90.74 | – | – | – | – | – |
| | AresB-18 | 91.90 | – | 93.02 | – | 1.12 | – |
| | AresB-34 | 92.71 | – | 93.62 | – | 0.91 | – |
| CIFAR-100[a] | AresB-10 | 69.45 | 91.70 | – | – | – | – |
| | AresB-18 | 73.01 | 92.57 | 75.61 | 93.05 | 2.60 | 0.48 |
| | AresB-34 | 74.73 | 93.25 | 76.76 | 93.37 | 2.03 | 0.12 |

Note:
[a] Full-precision (denoted as FP) counterparts of the AresB-18 and AresB-34 models are based on the evaluation results of the ResNet-18 and ResNet-34 models.

**Table 3 Comparison with other models containing baseline basic blocks on CIFAR-10 dataset.**

| Model | W/A[b] | Down sampling[c] | Top-1 (%)[d] | Storage (Mbits) | FLOPS (×10⁷) |
|-------|--------|------------------|--------------|-----------------|---------------|
| ResNet-18 (*He et al., 2016*)[a] | 32/32 | FPconv | 93.02 | 358 | 58.6 |
| XNOR-Net-18 (*Rastegari et al., 2016*) | 1/1 | Bconv | 89.83 | 12.9 | 1.41 |
| Bi-Real-Net-18 (*Liu et al., 2018*) | 1/1 | FPconv | 89.30 | 18.2 | 3.82 |
| AresB-18 | 1/1 | No conv | 91.90 | 12.7 | 1.36 |

Notes:
[a] A ResNet-18 model contains eight basic blocks.
[b] Terms *W* and *A* denote the precision of target weights and activation.
[c] Prefix *FP* and *B* mean the full-precision and binarized $1 \times 1$ convolutions, respectively.
[d] Top-1 accuracy indicates the final Top-1 test accuracy on the CIFAR-10 dataset.

additional accuracy enhancements. Compared with the experiments on the CIFAR-10 dataset, classification results were more improved on the CIFAR-100 by increasing the number of stacked blocks.

In Table 2, the final test accuracies are summarized, comparing with full-precision counterparts. On the CIFAR-10 dataset, the final test accuracies of the proposed model were slightly degraded over those of full-precision ResNet models. On the CIFAR-100 dataset, Top-1 accuracy of the AresB-18 model reached up to 73.01%, which degraded the classification accuracy by only 2.6% compared with the full-precision ResNet-18 model.

The efficiencies of the speedup and storage size were analyzed, assuming the combining factor between real-value and binary operations as $\frac{1}{64}$ (*Rastegari et al., 2016*). We assumed that the scaling in the BN layer and non-linear activation (e.g., ReLU) layer for one element increase FLOPs (floating-point operations per second) by one, respectively. The FLOPs of each convolution layer were calculated based on *Sagartesla (2020)*. The first convolution layer with RGB channels inputs and final fully-connected layer were operated on 32-bit full-precision data. Table 3 summarizes the comparisons with other models that contain the baseline basic blocks in terms of the speedup and storage size, where ResNet-18 (*He et al., 2016*), XNOR-Net-18 (*Rastegari et al., 2016*), Bi-Real-Net-18 (*Liu et al., 2018*) are compared. Because there was no $1 \times 1$ convolution in the downsampling, the FLOPs and storage size of our model became the smallest. Besides, TOP-1 test accuracy increased by 2.07% on the CIFAR-10 dataset. Compared with the theoretical speedup on

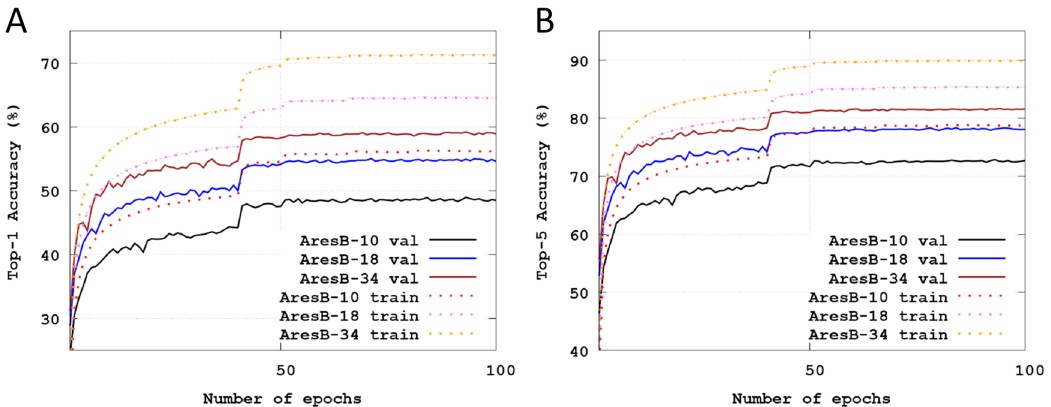

**Figure 4 Training and test classification accuracies across training epochs: (A) training and test Top-1 accuracy with the AresB-10, AresB-18 and AresB-34 models on the ImageNet dataset; (B) training and test Top-5 accuracy with the AresB-10, AresB-18, and AresB-34 models on the ImageNet dataset.**

ImageNet database in *Rastegari et al. (2016)*, the computation ratio of the first real-valued convolution layer was lower on the CIFAR dataset. The speedup over real-valued ResNet-18 was calculated by $\frac{\text{FLOPS(ResNet}-18)}{\text{FLOPS(AresB}-18)} \approx 44.73\times$ on the CIFAR dataset, which was smaller than $62.27\times$ speedup on the ImageNet database.

## Experiments on ImageNet dataset

The ImageNet dataset (*Russakovsky et al., 2015*) contains 1.2 million training and 50,000 validation color images classified into 1,000 categories. The image sizes were not fixed, so that images were resized into $256 \times 256$ images in the initial conversion. Then, each image was obtained by cropping the original image in the range of 0.466–0.875 and resized by $224 \times 224$. After applying the random horizontal flip, images were used in training. The random erasing in *Zhong et al. (2017)* was not applied in training, unlike the cases on the CIFAR dataset. Therefore, the random horizontal flip without the random erasing described in *He et al. (2016)* was adopted in the AresB-Net model. Additionally, the dropout layer was not adopted. When doing inference, $224 \times 224$ images were cropped from the center of original images without random flip.

We trained our AresB-10, AresB-18 and AresB-34 models from scratch for 100 epochs with a batch size of 256. For ADAM optimizer (*Kingma & Ba, 2014*) with β = (0.9, 0.999), momentum= 0.9 and weight decay= 1$e$−5. The initial learning rate $lr$ = 0.1 was decayed by multiplying 0.1 at (40, 50, 65, 80) epochs. Like ResNet (*He et al., 2016*), the AresB model started at the $7 \times 7$ convolutional layer with *channel* = 64 and *stride* = 2, followed by the $3 \times 3$ max-pooling layer with *stride* = 2. The test with the ImageNet validation dataset adopted only one random crop.

Figure 4 illustrates Top-1 and Top-5 classification accuracies across training epochs on the Imagenet datasets. The solid and dashed lines represent test and training accuracies, respectively. The validation images were used to test the trained model. Like the baselined pyramid structure in *He et al. (2016)*, as the number of stacked blocks increased, accuracies were enhanced. Compared with the test accuracies in Fig. 3, those of Fig. 4

**Table 4 Comparison with other models on ImageNet dataset(%).**

| Model[a] | Top-1 | Top-5 | Down sampling | From scratch[a] | Storage (Mbits) | FLOPS (×10⁸) |
|---|---|---|---|---|---|---|
| ResNet-18 (*Rastegari et al., 2016*) | 69.3% | 89.2% | BConv | Yes | 374.1 | 18.1 |
| XNOR-ResNet-18 (*Rastegari et al., 2016*) | 51.2% | 69.3% | FPConv | Yes | 33.7 | 1.67 |
| ABC-Net-res18 (*Lin, Zhao & Pan, 2017*) | 42.7% | 67.6% | BConv | No | 27.5 | 1.59 |
| Bi-Real-Net-18 (*Liu et al., 2018*) | 56.4% | 79.5% | FPConv | No | 33.6 | 1.63 |
| MoBiNet-k4 (*Phan et al., 2020*) | 54.4% | 77.5% | BConv | Yes | 36.8 | 0.52 |
| AresB-18 | 54.81 | 78.15 | No conv | Yes | 27.6 | 1.61 |

**Note:**
[a] When the model is trained from scratch, a pretrained model are not used in the weight initialization.

were varied smoothly. Finally, Top-1 and Top-5 accuracies were 48.51% and 72.72% with the AresB-10 model, 54.81% and 78.15% with the AresB-18 model and 58.46% and 81.22%, respectively.

Table 4 summarizes test accuracies and other important characteristics with XNOR-ResNet-18 (*Rastegari et al., 2016*), ABC-Net-res18 (*Lin, Zhao & Pan, 2017*), Bi-Real-Net-18 (*Liu et al., 2018*), and MoBiNet-k4 (*Phan et al., 2020*) comparable to our AresB-18 pyramid structure. In Table 4, *FPConv* and *BConv* denote floating-point and binarized 1 × 1 convolutions. Data in (*Liu et al., 2018*) assumed that XNOR-ResNet-18 adopted FPConv downsampling, which is referenced in Table 4. Our work outperformed other works except for Bi-Real-Net-18 that adopted the FPConv downsampling and needed more massive storage. Compared with results on CIFAR datasets, because the kernel size of the real-valued first convolution layer increased, the improvements in terms of FLOPS decreased. In addition, because all models started with real-valued 7 × 7 convolution layer and ended with real-valued fully connected layer for 1000 labels, the ratio of the reduced storage by removing the real-valued 1 × convolution also decreased. The MoBiNet-k4 model (*Phan et al., 2020*) can reduce FLOPS ≈ 3× over other BNN-based works. However, the removal of FPConv downsampling reduced storage size significantly over those of the MoBiNet-k4 model. Therefore, we conclude that AresB-Net can have merits in reducing storage size with acceptable test accuracies.

## Ablation studies

We conducted ablation studies with experimental results on the CIFAR datasets.

- **Effects of repeating channels in grouped convolution** We performed experiments to know the effects of the increasing number of channels in the AresB-Net model. Compared with the baseline ResNet (*He et al., 2016*), the basic block doubled the number of channels, but the grouped convolution maintained computational costs. By extending this idea, another experiment repeated channels and increased groups in the convolution by a factor of 2, multiplying the trainable shift and bias parameters in the BN layer with expanded channels. The experimental results enhanced overall test accuracies even though the computational costs in the grouped convolution maintained.

Compared with the original setup of the AresB-18 model, Top-5 test accuracies of the extended versions increased by 92.86% for 2× channels and 93.07% for 4× channels on the CIFAR-100 dataset, respectively.

- **Effects of the first BN layer in the second subblock** When omitting this BN layer, we experienced the gradient exploding in training on the CIFAR-100 dataset. Without this layer, several features from previous blocks can have direct effects on the filtering results, so that our version contained this BN layer per two subblocks.

- **Pooling layer in downsampling** As shown in Fig. 1B, the first subblock provided the downsampled shortcut with *stride* = 2, where 3 × 3 max-pooling layer with *padding* = 1 was adopted. Different types of pooling layers were applied to the AresB-18 model on the CIFAR-10 dataset. In addition to 2 × 2 max-pooling, 2 × 2 and 3 × 3 average pooling layers were adopted in modified versions. In these evaluations, the final Top-1 classification accuracies with different pooling layers were ranged in 91.54–91.58%, which were slightly degraded compared with the version using 3 × 3 max-pooling in the downsampling.

- **Data augmentation** An experiment was conducted to know how much the specific data augmentation affected the performance improvement. Without the random erasing (*Zhong et al., 2017*) in the data augmentation, the AresB-18 model on CIFAR-10 achieved 91.68% Top-1 final test accuracy. Compared with the final classification result without the random erasing (91.90%), slight accuracy enhancements were shown with this specific data augmentation technique. Therefore, it was expected that this augmentation technique could improve the performance. But the increase was not significant, which means that the performance enhancement was mainly caused by the proposed AresB-Net model.

## CONCLUSION

The proposed network model achieves significant test accuracy improvements with reduced costs, by expanding output channels and applying shuffled grouped convolutions. The advantages of existing network blocks are combined along with the convenience of making the pyramid structure. For apple-to-apple comparisons, we focused on the basic block structure, so that we did not apply any specific training schemes and weight initialization. In addition, our model did not consider trainable parameters for scaling convolution outputs (*Bulat & Tzimiropoulos, 2019*), tuning binary activation (*Wang et al., 2020*; *Liu et al., 2020*), PReLU (*He et al., 2015*). We definitely expect that there is no difficulty in applying state-of-the-art training schemes and tuning methods to our model. When adopting the basic training optimization and training from scratch, our model can achieve acceptable performance for the CIFAR and ImageNet datasets and reduce hardware costs by removing 1 × 1 downsampling. Notably, this proposed model provides significant benefits in terms of storage size and speedup on CIFAR datasets. By removing the intervention of the real-valued 1 × 1 convolution in the middle of operating a BNN model, BNN's operating steps become more simple. Most of all, it is concluded that the proposed model can provide good classification results with low computational costs.

### Funding

The author received no funding for this work.

### Competing Interests

The author declares that they have no competing interests.

### Author Contributions

- HyunJin Kim conceived and designed the experiments, performed the experiments, analyzed the data, performed the computation work, prepared figures and/or tables, authored or reviewed drafts of the paper, and approved the final draft.

### Data Availability

Code is available at GitHub: https://github.com/analog75/aresb/

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
