# Peer review of "AresB-Net: accurate residual binarized neural networks using shortcut concatenation and shuffled grouped convolution"

_PeerJ Computer Science, doi:10.7717/peerj-cs.454_

## Round 0.1 · original submission · Major Revisions

Overall, the presentation should be improved as all reviewers indicated. It is not easy to understand this manuscript.

As reviewer #1 indicated, authors are encouraged to provide detailed explanations on terminology and experiments. Experimental results should be elaborated. Please provide more detailed results.

Reviewer 1 ·

Basic reporting

no comment

Experimental design

no comment

Validity of the findings

no comment

Additional comments

The paper proposed AresB-Net which achieves better accuracy with reduced costs. This network consists of basic blocks where the real-valued shortcut per each binarized convolution layer by adding the shortcut and concatenating it to output channels is added, and then the expanded channels are shuffled and split for the next grouped convolution. The experiments are conducted on CIFAR10, CIFAR100, and ImageNet with the comparison of other algorithms, and the proposed method does have less storage size with acceptable test accuracies. It seems that there exist some points that can be considered as a novel work, however, there are some confusing throughout the texts. The major comments are as follows:

(1) (Page 4 line 127-159) Even though I have sufficient background in CNN and BNN research, it has been very difficult for me to understand what exactly was done in the novel basic block. To make this work accessible, it would be necessary to carefully rework the presentation. For example, the output sizes of the basic block are explained thoroughly, however, it is recommended to add a table describing the output sizes of the basic block to make it more readily understandable.
(2) Please provide some more explanation or revise the terminologies. (line141 subblocki, line142 1/n Symbols *)
(3) Abbreviation of BinConv3x3 layer should be properly represented (Page 4 line 131).
(4) Please enrich the manuscript with references or detailed experimental results for these; The evaluation using LeakyReLU having better performances than those using normal ReLU (Page 4 line 151-154) and the better classification accuracy with BN layer every two subblocks (Page 4 line 158)
(5) In page 7 line 249, it is indicated that the speedup over real-valued NesNet-18 on CIFAR dataset can be over 44%. Please clarify.
(6) Please provide the correct reference for the calculation method of the FLOPS of convolution layers (Page 6 line 241).
(7) In Section Ablation studies, four case studies are described. However, experimental results are not really mentioned thoroughly. Please provide some more detailed results. For example, it is mentioned that slight accuracy enhancements were shown with the random erasing for data augmentation. Please provide more detailed experimental results and why you chose this method for data augmentation.

Reviewer 2 ·

Basic reporting

- For me, the section, ARESB-NET: ACCURATE RESIDUAL BINARIZED NEURAL NETWORK MODEL, is quite not easy to understand. I think it would be better to rearrange the contents.

- Also it would be better to mention the reason for using the block for stride=2 and why the number of output channels is doubled.

Experimental design

- the results verify that the proposed network achieve compatible performance while reduces computational costs

- It would be better to add some experiments that can show the validity of block design (ex) comparison between using left and right group for skip connection, and accuracy and required storage when adding 1x1 convolution)

- Please be clear whether the experiments were conducted under the same conditions(using the same data augmentation technique?)

- Is the storage-saving effect coming from the absence of 1x1 convolution is enough to account for the reduction in accuracy? (the storage is not significantly different from Bi-RealNet-18)

Validity of the findings

- I recommend analyzing more the reason why the proposed network shows good performance without 1x1 convolution

---

## Round 0.2 · accepted · Accept

Thanks for revising the paper in accordance with the reviewers' comments. It is now can be published.

Reviewer 1 ·

Basic reporting

.

Experimental design

.

Validity of the findings

.

Additional comments

The manuscript is well-revised and the issues raised by the reviewer has been clarified. Therefore, the paper is now acceptable for publication.

Reviewer 2 ·

Basic reporting

In this paper, the author proposes AresB-Net which is a residual binarized CNN with improved accuracy. For this, a pyramid structure without downsampling convolution is suggested. And the performance
of AresB-Net is verified through several experiments.

- After the revision, the readability has increased and the structure has become more clear.

- Also, the proposed network is clearly explained in detail and I could fully understand the method and its goals.

- The author provides proper figures and tables that call well explain the proposed model.

Experimental design

- The results verify that the proposed network achieve compatible performance while reduces computational costs

- The experimental environment, conditions, parameters, and the datasets the author uses are described well.

- The author properly designed the experiments. And the storage-saving effect and accuracy improvement effect of the proposed model are shown through the experiments.

Validity of the findings

In this paper, the author proposes AresB-Net, and explains the structure and characteristics of the proposed network in detail, The performance of this network is verified through various experiments on several datasets.

Additional comments

Thank you for your hard work and for giving a sincere response.